# Synthesis of Diazacyclic and Triazacyclic Small-Molecule Libraries Using Vicinal Chiral Diamines Generated from Modified Short Peptides and Their Application for Drug Discovery

**DOI:** 10.3390/ph17121566

**Published:** 2024-11-22

**Authors:** Mukund P. Tantak, Ramanjaneyulu Rayala, Prakash Chaudhari, Chhanda C. Danta, Adel Nefzi

**Affiliations:** 1Herbert Wertheim College of Medicine, Center for Translational Science, Florida International University, Port Saint Lucie, FL 34987, USA; mukundtantak.2009@gmail.com (M.P.T.); rraya002@fiu.edu (R.R.); pchaudha@fiu.edu (P.C.); cdanta@fiu.edu (C.C.D.); 2Department of Chemistry and Biochemistry, College of Arts, Sciences & Education, Florida International University, Miami, FL 33199, USA

**Keywords:** vicinal amines, combinatorial chemistry, nitrogen-containing scaffold, diazacyclic compounds, triazacyclic compounds, natural product-like compound libraries, drug discovery, heterocyclic compounds, peptidomimetics, polyamines, solid-phase organic synthesis, small-molecule libraries, diversity-oriented synthesis

## Abstract

Small-molecule probes are powerful tools for studying biological systems and can serve as lead compounds for developing new therapeutics. Especially, nitrogen heterocycles are of considerable importance in the pharmaceutical field. These compounds are found in numerous bioactive structures. Their synthesis often requires several steps or the use of functionalized starting materials. This review describes the use of vicinal diamines generated from modified short peptides to access substituted diaza- and triazacyclic compounds. Small-molecule diaza- and triazacyclic compounds with different substitution patterns and embedded in various molecular frameworks constitute important structure classes in the search for bioactivity. The compounds are designed to follow known drug likeness rules, including “Lipinski’s Rule of Five”. The screening of diazacyclic and traizacyclic libraries has shown the utility of these classes of compounds for the de novo identification of highly active compounds, including antimalarials, antimicrobial compounds, antifibrotic compounds, potent analgesics, and antitumor agents. Examples of the synthesis of diazacyclic and triazacyclic small-molecule libraries from vicinal chiral polyamines generated from modified short peptides and their application for the identification of highly active compounds are described.

## 1. Introduction

The development of synthesis routes for new families of Aza heterocyclic compounds currently represents a significant challenge in medicinal chemistry and chemical biology [1,2,3,4,5]. The increase in the structural complexity of “small molecules”, by incorporation of sp^3^-hybridized and asymmetric carbons, makes it possible to induce a better exploration of “chemical space” [6,7,8,9], in terms of three-dimensional structure, as well as an increase in their specificity and their bioavailability. Vicinal amines provide useful building blocks in organic synthesis and play an important role in natural products, organic synthesis, drug design, and the development of synthetic approaches since the generation of vicinal diamines continues to attract the interest of chemists and medicinal chemists for their broad biological and synthetic applications [10,11,12]. Many approaches have been reported for the synthesis of vicinal diamines [10,12,13,14,15], but the reduction in polyamide bonds remains the most direct and most diverse approach for the generation of substituted chiral vicinal amines [16,17,18,19,20,21]. Using the large diversity afforded by the solid-phase synthesis of peptide libraries [22,23,24,25] as well as the optimized chemistry that provides peptides and peptidomimetics in good synthetic purity, an area of active research of our group has been the development of synthetic approaches to chemically transform resin-bound peptide and peptidomimetic libraries to nitrogen-based acyclic and heterocyclic compounds [24,26,27,28,29]. Known as the “libraries from libraries” technique [29,30], the chemical modification of polyamide libraries leads to a variety of peptidomimetic libraries as well as nitrogen-based acyclic and heterocyclic low-molecular-weight small-molecule libraries with different physicochemical and biological properties compared to the original peptide libraries. Examples of these chemical modifications include the amide peralkylation and/or exhaustive reduction in amides to generate the corresponding peralkylated polyamides and polyamines [29,30]. In this paper, examples of the diversity-oriented synthesis of a variety of diazacyclic and triazacyclic small-molecule libraries from vicinal chiral diamines generated from resin-bound short peptides and their screening results toward the identification of hit compounds are described.

The development of new strategies for innovative peptidomimetic design led to creating different unique metabolically stable nitrogen-based pharmacophores that mimic the bioactive conformation of the peptide by adopting different conformations in space and distributing the side chains of amino acids for biological activity [31,32,33,34]. The parallel synthesis of novel heterocyclic compounds derived from resin-bound vicinal diamines derived from reduced polyamides has been the subject of intense research interest in our group [26,29,35]. Additionally, the development of strategies for making fused and tethered heterocyclics has allowed access to new pharmacophores, which, through hydrophobic interactions, hydrogen bonding, or complexation, subsequently form well-defined new structure–activity relationship (SAR) insights, ultimately leading to new active compounds with improved stability and oral bioavailability. Importantly, this new approach could have widespread applicability to a variety of ligand synthesis efforts.

We used the “libraries from libraries” technique [16,29,30] to transform the peptide skeleton into heterocycles while maintaining the side chains of the amino acids that are responsible for interactions with the different receptors. This approach allows the transformation of metabolically unstable compounds into compounds that possess drug-like properties [36]. Entirely diverse new small-molecule libraries are generated by the chemical modification of existing libraries, providing different compounds with novel physical, chemical, and biological properties.

## 2. Synthesis of Diazacyclic and Triazacyclic Compounds from Vicinal Chiral Diamines

The pharmaceutical use of peptides faces limits, whether linked to the low capacity of peptides to cross membrane barriers, their short half-life, lack of oral bioavailability, and rapid hydrolysis [37,38,39,40]. Therefore, the development of methodologies for the modification of peptide libraries is highly desirable. One of the most interesting modifications to peptide backbone is the generation of vicinal chiral polyamines following the exhaustive reduction in amide bonds [16,41]. The “libraries from libraries” approach has been successfully employed for the generation of a variety of diazacyclic and triazacyclic small-molecule libraries from vicinal chiral diamines generated from modified short peptides [26,28,29,30,35].

As subsequently reported, the typical reaction conditions for the exhaustive reduction in amide bonds consist of the reduction in resin-bound polyamides with the BH_3_-THF complex [16,41]. We and other groups reported that the borane reduction in amide bonds is free of racemization by comparing the relative absorbances of different pairs of diastereomers that do not coelute [41,42].

Figure 1 illustrates an example of the generation of a variety of diazacyclic heterocyclic peptidomimetics from resin-bound *C*-terminal *N*-alkylated acylated dipeptides. The treatment of the vicinal triamines containing one tertiary amine and two secondary amines with different bifunctional reagents, such as carbonyl diimidazole, thiocarbonyl diimidazole, cyanogen bromide, oxalyldiimidazole, chloroacetaldehyde, malonyl chloride, and *N*-chlorocarbonyl isocyanate, produced, following the cleavage of the solid support, the corresponding trisubstituted cyclic urea, cyclic thiourea, cyclic guanidine, diketopiperazine, piperazine, triazepinediones, diazepinediones, and diketopiperazines, in a good yield and with good purity [27,43,44,45]. This is an example of a broader approach to the solid-phase diversity-oriented synthesis of diverse diazacyclic compounds from vicinal diamines using modified peptides as starting materials.

Using the Houghten’s “tea bag” method of parallel synthesis [46], and following the strategy outlined in Figure 1 [47], different diazacyclic combinatorial libraries were prepared in the positional scanning (PS) format, each containing 118,400 compounds of the aforementioned heterocycles. The concept of positional scanning libraries (PSLs) and their use for the identification of active peptides and small molecules have been previously described and reviewed in many reports [23,24,48,49].

A successful story of the use of positional scanning mixture-based libraries for drug discovery is the recent discovery of a tetrapeptide kappa opioid receptor (KOR) agonist that targets the body’s peripheral nervous system. Difelikefalin, an all D-amino acid tetrapeptide, has been approved by the FDA for the treatment of moderate-to-severe pruritus in hemodialysis patients [50,51]. The original selective kappa agonist tetrapeptide was originally identified from a tetrapeptide positional scanning library consisting of 6.25 million tetrapeptides [52,53].

Figure 1 illustrates the concept of the preparation and deconvolution of the library of 1,5-disubstituted acylated 2-amino-4,5-dihydroimidazoles for the identification of a novel class of retinoic acid receptor-related orphan receptor (ROR). The library containing three positions of diversity was prepared from resin-bound vicinal diamines following the strategy described in Figure 2.

As outlined in Figure 1, the dihydroimidazole library, having three positions of diversity (R^1^, R^2^, and R^3^), consists of three separate sub-libraries, each containing one defined position (R) and two non-defined mixture positions (X). The figure also explains the strategy to deconvolute an active library by the determination of the most active groups at each of the three positions of this library directly from the screening results. The screening of the three sub-libraries leads to the most active groups of each variable position in the 1,5-disubstituted acylated 2-amino-4,5-dihydroimidazoles library. The deconvolution of the library identified four active mixtures with a defined position R^1^, three active mixtures with a defined position R^2^, and four active mixtures with a defined R^3^ with a good inhibition activity against ROR. The solid-phase parallel synthesis and screening of the 48 (4 × 3 × 4) individual compounds representing the combination of active mixtures with defined R^1^, R^2^, and R^3^ using the strategy outlined in Figure 1 led to the identification of individual compounds with a good activity against RORγ (3.3 µM IC_50_) and almost a two-fold selectivity toward this receptor isoform, with 5.3 and 5.8 µM IC_50_ against RORα and RORβ cells, respectively.

Using the same concept, previous results from our laboratory have shown the utility of the new diazacyclic libraries for the identification of highly active compounds (Figure 2). The screening of the positional scanning library (PSL) of cyclic urea identified active antifungal trisubstituted cyclic urea with a minimum inhibitory concentration (MIC) ranging from 64 to 125 µg/mL [54]. Potent antimicrobial trisubstituted cyclic thiourea inhibitors of tyrosine recombinases and Holliday junction (HJ)-resolving enzymes were identified following the screening of the cyclic thiourea library [55,56]. This inhibitor binds specifically to protein-free HJs and can inhibit HJ resolution by RecG helicase.

The trisubstituted chiral 1,2,4-trisubstituted piperazines library was screened against the *Mycobacterium tuberculosis* strain H37Rv. The activity of the compounds against five drug-resistant Mycobacterium tuberculosis (*Mtb*) strains-2 isoniazid-resistant (INH-R1 and INH-R2), a fluoroquinolone-resistant strain (FQ-R1), and two rifampicin-resistant strains (RIF-R1 and RIF-R2) were evaluated by the determination of the MIC based on an optical density (OD) readout. Compounds with IC_50_ in the range of 17–25 µM were identified.

The diazacyclic *N*-methylated 1,3,4-trisubstituted piperazine and *N*-benzylated 1,3,4-trisubstituted piperazine were screened against the three opioid receptors mu, kappa, and delta. The *N*-methyl piperazine library was found to be more active than the *N*-benzylated piperazine library for both mu and kappa receptors [57]. The screening and deconvolution of the *N*-benzylated piperazine library toward the identification of polyamine transport inhibitors identified active leucine and methionine depletion agents [58]. The screening of cyclic guanidine-linked sulfonamides library identified several compounds with sub-micromolar activity against LMTK3 kinase as a potential target for anticancer drug development [59].

On the other hand, the screening of the trisubstituted piperazine and trisubstituted 2,3-diketopierazine libraries led to the rapid identification of antagonists of the nuclear retinoic acid receptor-related orphan receptor gamma (RORγ) [60]. A novel series of retinoic acid receptor-related orphan receptor (ROR) inhibitors were identified following the screening of the 1,5-disubstituted acylated 2-amino-4,5-dihydroimidazole library generated from the acylation of resin-bound cyclic guanidines [61].

Continuing with the synthesis of diazacyclic compounds from vicinal diamines, an efficient approach toward the solid-phase synthesis of monoketopiperazine ring systems from resin-bound acylated amino acids was developed (Figure 3). Following the reduction in resin-bound acylated amino acids, the vicinal diamines were treated with an excess of bromoacetic acid in the presence of the coupling reagent diisopropylcarbodiimide (DIC) and diisopropylethylamine (DIEA). The acylation of one of the secondary amines with bromoacetic acid was immediately followed by an in situ intramolecular substitution of the bromo group with the other vicinal amine, resulting in the formation of the six-membered monoketo-piperazine. The cleavage of solid support produced the final desired products in a good yield and with good purity.

## 3. Synthesis of Fused Diazacyclic Compounds from Vicinal Polyamines

Starting with resin-bound vicinal triamine containing three available secondary amines derived from reduced resin-bound *N*-acylated dipeptides, a similar approach was used for the solid-phase synthesis of fused bicyclic guanidines. The treatment of resin-bound vicinal triamine with thiocarbonyl diimidazole in the presence of mercury acetate (Hg(OAc)_2_) led, following the cleavage of the solid support, to the desired bicyclic guanidines (Figure 4) [16].

The screening of the 100,000 bicyclic guanidine library, generated in a positional scanning format in a variety of assays, led to the identification of highly active compounds, including potent antifungal activities against *Candida albicans* and *Cryptococcus neoformans* [62], ligands with a high binding affinity against the kappa receptor [57] and prohormone convertase 2 inhibitors (Figure 3) [63].

Similarly, a well-designed approach for the solid-phase parallel synthesis of trisubstituted azoniaspiro diazacyclic compounds from resin-bound vicinal triamines was reported [64]. The target compounds, the 1,8,9-trisubstituted 10-oxo-3,9-diaza-6-azoniaspiro [5.5]undecanes, were obtained starting with reduced resin-bound dipeptides, as illustrated in Figure 5. The azoniaspiro cation was obtained following an intramolecular attack of a tertiary amino group on the pendent bromoacetyl group.

The solid-phase parallel synthesis of *N*-alkyl-4,5,7,8-tetrahydro-3H-imidazo [1,2-a][1,3,5]triazepin-2-amines starting with resin-bound vicinal triamines was reported [65,66,67]. As illustrated in Figure 6, the key step involved the use of bifunctional thiocarbonyldiimidazole for the cyclization of an amino with a guanidino group, followed by a subsequent transformation of the generated thiourea moiety to a substituted amino triazepine in the presence of mercury acetate and various amines. The desired products were obtained following the cleavage of the resin in good yields and with good-to-moderate purities, depending on the nature of the building blocks used.

Starting with resin-bound acylated tripeptides and following the reduction in the amide groups and the cleavage of the solid support, the generated crude tetraamines were then separately treated with diethyl malonoimidate dihydrochloride and DIEA in anhydrous DMF (Figure 7) [68]. The solutions were stirred in separate vials at 85 °C for 48 h to produce the subsequent evaporation of the solvent’s oily solutions. The crude compounds were then dissolved in acetic acid, frozen, and lyophilized, and the obtained white powders were purified using semi-preparative high-performance liquid chromatography (HPLC). The desired fused diazacyclic diimidazodiazepines were obtained in a good yield and with good purity.

The diimidazodiazepine compounds were screened in vitro against the three-opioid receptor μ(MOR), δ (DOR), and κ (KOR). The in vitro screening binding assays provided a variable affinity for opioid receptors (Figure 4). The identified lead compound, namely 2065-14 [68], was then tested in vivo and found to be a peripherally restricted mixed kappa–delta agonist. The compound is now in clinical trials (as CAV1001) by Caventure Drug Discovery, Inc. for the development of opioids free of the common opioid side effects while retaining efficacy in neuropathic pain states, inflammatory pain states, and in visceral pain states [69,70,71].

The central (i.c.v.), intraperitoneal (i.p.), or oral (p.o.) administration of 2065-14 produced dose-dependent, opioid-receptor-mediated antinociception in the mouse 55 °C warm-water tail-withdrawal assay. No traces of the compound were found in the brain up to 90 min later, suggesting no BBB penetration and presumably peripherally restricted activity [68].

## 4. Synthesis of Bis-Diazacyclic Compounds from Alternated Vicinal Diamines

Continuing with our efforts toward the generation of heterocyclic peptidomimetics from modified resin-bound peptides, we reported the synthesis of different classes of bis diazacyclic compounds [18].

We designed and performed the solid-phase parallel synthesis of different pyrrolidine bis diazacyclic libraries from resin-bound acylated tetrapeptides. As outlined in Figure 8, Boc-proline was coupled to resin-bound amino acids (position of diversity R^1^), followed by the subsequent coupling of two Boc-amino acids (positions of diversities R^2^ and R^3^). The *N*-terminal was *N*-acylated with different commercially available carboxylic acids (position of diversity R^4^). The exhaustive reduction in the resin-bound polyamides led to resin-bound vicinal pentaamines, having two pairs of vicinal secondary amines separated by a pyrrolidine ring. Using the previously described “libraries from libraries” approach, and different bifunctional reagents such as thiocarbonyldiimidazole, cyanogen bromide, and oxalyldiimidazole, we obtained, following the cleavage of the solid support, the corresponding bis diazacyclic libraries, pyrrolidine bis-cyclic thiourea, pyrrolidine- bis-cyclic guanidines, and pyrrolidinebis-diketopiperazine.

Using 26 different commercially available amino acids at each of the three positions of diversity and 42 carboxylic acids at the fourth position, each library has a total of 738,192 compounds arranged in the positional scanning format. The phenotypic screening of these bis diazacyclic libraries in a variety of assays identified potent compounds.

Similarly, we designed and developed a strategy for the solid-phase synthesis of bis-heterocyclic libraries from resin-bound orthogonally protected lysine (Figure 9). We designed and prepared resin-bound orthogonally protected Fmoc- Lys (Boc)-OH for the generation of poly vicinal amines for the synthesis of a variety of “bis”-diazacyclic compounds. We used the Fmoc/Boc orthogonality [72], subsequent amino acid coupling and acylation, and exhaustive reduction in the amide bonds, the generated resin-bound separated pairs of vicinal diamines were treated with carbonyldiimidazole, thiocarbonyldiimidazole, cyanogen bromide, and oxalyldiimidazole to yield the energetically favored five and six-membered rings, corresponding to bis-cyclic ureas, bis-cyclic thioureas, bis-cyclic guanidines, bis-diketopiperazines, and bis-piperazines. Using 42 possible substitutions for R^1^, 26 possible substitutions for R^2^, and 42 possible substitutions for R^3^, a total of 45,864 compounds per library were prepared in the positional scanning format.

Similarly, as outlined in Figure 10, resin-bound dipeptides derived from Boc/Fmoc orthogonally protected diamino acids, such as lysine or ornithine, were also explored for the synthesis of a variety of imidazoline-tethered diazacyclic compounds [73,74,75]. Following reduction, the primary amine of the resulting tetraamines was selectively protected with the 1-(4,4-dimethyl-2,6-dioxocyclohex-1-ylidene)ethyl (Dde)-protecting group [76]. The free secondary vicinal diamines were then treated with different bifunctional reagents (COIm)_2_, COIm_2_, and CSIm_2_). Following Dde removal, the primary amine was selectively *N*-acylated and dehydratively cyclized in the presence of POCl_3_ in anhydrous dioxane at 100 °C to produce, following dehydration, the corresponding imidazoline-tethered diazacyclic compounds in a good yield and with good purity. These procedures were extended to prepare different mixture-based combinatorial libraries.

We extended the above-mentioned strategies to transform vicinal tetramines derived from resin-bound tripeptides [19]. The generated resin-bound tetraamines contained three secondary amines and one terminal primary amine (Figure 11). The resin-bound tetra-amine was then treated with bifunctional reagents to produce, following the cleavage of solid support, the corresponding bis-diazacyclic compounds. The primary amine reacted first with the bifunctional reagent, followed by the kinetically favorable intermolecular cyclization with the adjacent secondary vicinal amine to yield the corresponding diazacyclic ring. The remaining secondary vicinal amines then further reacted with the excess of the same bifunctional reagent to yield the second diazacyclic ring. Based on this optimization, we observed that all the reactions under lower concentrations of the bifunctional precursors yielded the desired bis-diazacyclic compounds with purities greater than 80%.

In collaboration with a large number of scientists, the screening of the bis-diazacyclic libraries has shown the utility of these classes of compounds for the de novo identification of highly active compounds (Figure 5), including antimicrobial compounds [55,77,78], antimalarial [79], antitubercular [18], potent analgesic [80,81], novel neurite outgrowth promoters [75], Gyrase inhibitors [82], Melanocortin-3 receptor (MC3R) agonists [83], and novel antitumor agents [84,85].

Examples of identified active compounds include the identification of potent low-liability antinociceptive bis cyclic guanidines from the direct in vivo screening of a pyrrolidine bis-cyclic guanidine library, totaling 738,192 compounds. The identified compounds demonstrated a dose-dependent antinociceptive effect three to five times greater than morphine that was antagonized by mu or mu and kappa opioid receptor-selective antagonists. The compounds produced significant respiratory depression, hyperlocomotion, or conditioned place preference [81].

The screening of the pyrrolidine bis-cyclic guanidine library against the chloroquine (CHQ)-resistant Pf Dd2 strain led to the rapid identification of antiplasmodial compounds, which exhibited interesting antimalarial activity [79]. The same library was screened against drug-resistant Gram-positive pathogens [77]. At least 20 individual compounds were bactericidal for MRSA at ≤2.5 lg/mL, with a subset of these compounds showing bactericidal activities (≤10 lg/mL) against the other species tested.

The in vitro screening of bis-cyclic guanidine compounds derived from vicinal triamines in radioligand competition binding assays identified compounds with a variable affinity for the mu-opioid receptor (MOR), delta-opioid receptor (DOR), and kappa-opioid receptor (KOR) across the series, with compound 1968-22 displaying a good affinity for all three receptors. The central administration of compound 1968-22 did not produce locomotor effects, significant respiratory depression, or conditioned place preference or aversion. The data suggest these bis-cyclic guanidine with multifunctional opioid receptor activity may hold potential as new analgesics with fewer liabilities of use [80].

The screening of different bis diazacyclic compounds against the Mycobacterium tuberculosis strain H37Rv have led to the identification of submicromolar unique antitubercular hits. Highly active compounds with 90–100% inhibition were identified. Some of the identified compounds were more active than the existing drug ethambutol [18].

## 5. Synthesis of Oligodiazacyclic Compounds

By alternating two secondary amides and one tertiary amide, we used the same approach to generate pairs of vicinal diamines separated by a pyrrolidine ring derived from proline. As outlined in Figure 12, a variety of acylated heptapeptides were prepared using standard solid phase Boc strategy [86]. Following the reduction in the amide groups, the treatment of the resin-bound pairs of vicinal diamines with cyanogen bromide led, following the cleavage of the resin, to the corresponding polycyclic-guanidines (Figure 12) in a good yield and with high purity [82].

Similarly, we developed a strategy in which piperazine was incorporated as a source of tertiary amines for the synthesis of polycyclic guanidines (Figure 13). The strategy consists of subsequent coupling of two amino acids followed by acylation with bromoacetic acid and the substitution of the bromo group with Boc-piperazine. Following the reduction in the amide groups, the corresponding pairs of vicinal secondary amines separated by piperazine were treated with BrCN to generate, after the cleavage of the solid support, the corresponding piperazine-based polycyclic guanidines. The screening of the polycyclic guanidine compounds against *Mycobacterium tuberculosis* DNA gyrase inhibitors identified active compounds that did not inhibit human DNA topoisomerase IIα and topoisomerase I [82].

Continuing with the same approach on the functionalization of poly vicinal diamines, we developed a straightforward strategy for the generation of a variety of new macrocyclic-containing diazacyclic compounds derived from cyclic poly vicinal diamines (Figure 14) [87]. Following the reduction in the amide groups, the treatment of the predesigned separated pairs of vicinal secondary amines with different bifunctional reagents, such as oxalyldiimidazole and thiocarbonyldiimidazole, led, after the cleavage of the solid support, to the corresponding macrocyclic oligo diazacyclic compounds in good yields and with good purities.

## 6. Conclusions

The large number of molecules synthesized, the speed of their preparation, and the economy of the purification and product characterization steps, compared to the classic “drug discovery” method, are the reasons for the success of the “tea bag approach” [46], which has permitted the use of the “libraries from libraries” concept for the parallel synthesis of large numbers of individual compounds as well as mixture-based combinatorial libraries.

Using resin-bound vicinal polyamines, the synthetic strategies and combinatorial techniques presented above were applied to generate diverse, large numbers of diazacyclic and triazacyclic compounds for biological screening and led to the identification of active compounds for potential development as therapeutics and/or diagnostic agents.

## Data Availability

Not applicable.

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
