# Peer review of "Synthesis of Diazacyclic and Triazacyclic Small-Molecule Libraries Using Vicinal Chiral Diamines Generated from Modified Short Peptides and Their Application for Drug Discovery"

_pharmaceuticals, 2024, doi:10.3390/ph17121566_

Round 1
Reviewer 1 Report
Comments and Suggestions for Authors
The present review manuscript by Nefzi group presents the titled study, that is, synthetic methodology for substituted diaza- and triaza-cyclic compounds. Various these products could be provided by “libraries from libraries” approach mainly as shown in author’s longstanding works (11 original reports). Also, “tea-bag approach” introduced by Dr. Houghten could be applied to the research and development for the project.
Stereochemistry of stereogenic centers is the highlighted topic in synthetic organic chemistry and the chiral discrimination of pharmaceuticals. Accordingly, referring to the original reports, please describe this important subject with the information of the absolute configuration (R/S or D/L amino acids). When the stereochemistry is not strictly confirmed in the equation, asterisks “*” on chiral carbons should be provided in all Schemes and some Figures. Separated equations may provide the accessible information to readers.
This extensive review manuscript is worthy of note for reasonable members engaged in this area, the reviewer recommends the publication in the journal after major revisions described below.
A number of inappropriate descriptions should be revised along with “comments and suggestions” below.
<General comments and suggestions>
1. Title; Two “from” seems to be inappropriate for style of the review. More concise phrase is recommended.
2. Some “C-”, “N-”, “R”, “S” in the text should be carefully checked and altered to “italic style”
3. For some long sections, appropriate line breaks will help for smooth reading.
4. Titles of Scheme 4 and others; The first words should be capitalized such as synthesis of → Synthesis of (consistency).
5. Upper script style is generally used for R1, R2, …, in order to discriminate formulas such as Cn, Nn, etc (lower script). in organic chemistry. But, the manner is not compulsory.
6. Heading of Scheme 3 (p. 13) is incorrect and should be changed to Scheme 10. Sequentially, altered the number.
7. In the text, “good yield and purity” and “good yield and purities” are mixed. “good yield with sufficient chemical purity” may be suitable. Arrange this issue.
8. All schemes and text sentences; CNBr → BrCN or BrC≡N, COIm2 → Im2CO or Im2C=O, CSIm2 → Im2CS or Im2C=S, (COIm)2 → (ImCO)2 or (ImC=O)2 are appropriate and easy to understand in organic chemistry. But, the manner is not compulsory.
9. In Schemes 8, 9, 11 (not Scheme 4), insertion of “A”, “B”, “C”, … in the arrow equations will support readers’ easier comprehension. In accord with this presentation, it is necessary for the reference in text sentences. But, the manner is not compulsory.
<Comments and suggestions>
1) Affiliation of C. C. Danta should be provided.
2) P. 2, lines 5 & 7; “Vicinal amine” and “vicinal diamine” are repetitive. Please check it.
3) P. 4; The abbreviation of “SCLs” (“DCLs” ?) and “KOR” should be explained for readers not specialized in the area.
4) P. 4, line 7; The all D-amino acids tetrapeptide (Difelikefalin) → Difelikefalin, a tetrapeptide with all D-amino acids?
5) P. 7, line 2; Fig 2 → Figure 2 (consistency)
6) P. 7; The abbreviation of “HJs”, “HJ”, “RecG”, and “Mtb” should be explained for readers not specialized in the area.
7) P. 8, line 3; from → form
8) P. 8, line ↑3; Fig 3 → Figure 3 (consistency)
9) P. 9, line 9; “Scheme 4” should be to move to line 2 after “was after developed” for readers’ easier comprehension.
10) P. 9, line 6; the vicinal amine → the other vicinal amine
11) P. 9, line ↑3; -3,9- diaza → -3,9-diaza
12) P. 9, line ↑2; dipeptides. As illustrated in Scheme 5, → dipeptides as illustrated in Scheme 5,
13) P. 9, line ↑1; a tertiary nitrogen → a tertiary amino group (consistency) bromoacetyl. → bromoacetyl group.
14) Scheme 5; (COIm)2 → Im2C=O BrCH2COOH → BrCH2COOH
15) Scheme 5; Concerning the formula “N+”, what is the counter anion? Br̶ or Xr̶ ? In order to complete the formula, please provide it.
16) P. 10, line 3; cyclization of an amino and guanidino → cyclization of an amino group with guanidino?
17) P. 10, line 4; guanidine → triazepine?
18) P. 10, line 6; [64-66] should be move to line 2 after “was reported” for readers’ easier comprehension.
19) P. 10, lines 8, 9; Two “from vicinal tetraamines” are repetitive in a short sentence. Arrange it.
20) P. 10, lines 8, 9; Two “Starting from” are repetitive. Arrange it.
21) P. 10, line 11; Concerning “separately treated”, two regioisomers or stereoisomers were produced? If so, please provide the isomer formula in Scheme 7.
22) P. 10, line 11; were then separately treated in DMF in the presence of ### and DIEA → were then separately treated with ### and DIEA in DMF (as a standard description of synthetic chemistry)
23) P. 10, line 12; Concerning “separately stirred in separate vials”, more a smart description should be needed.
24) P. 10, line ↑1; (Fig. 4) → Figure 4 (consistency)
25) P. 11, line 1; Concerning “2065-14”, is this optimized or most active compound? And the correlation of other three compounds should be briefly addressed.
26) P. 11, line 8; “[18]” should be moved after “… compounds” in line 6 for readers’ easier comprehension.
27) P. 11, line ↑7; The N-terminal was N-acylated with → The N-terminal position was acylated to afford the N- precursor with
28) P. 11, line ↑3; thiocarbonylimidazole → ImC=S
29) P. 12, line 7; (Scheme 9) should be moved after “was performed” for readers’ easier comprehension.
30) P. 13, line 1; Similarly, and as outlined in Scheme 10, → Similarly, as outlined in Scheme 10,
31) P. 13, line 4; protecting group → group (repetition)
32) Scheme 10 (not 3); What is the condensation reagent in the first equation? POCl3? “Acylation (R2COOH / POCl3)” in all equation is more appropriate in the organic synthesis style.
33) P. 14, line 2; tripeptide. → tripeptide [73?].
34) P. 14, line 5; energetically → kinetically
35) P. 14, line 6; Two “adjacent” is repetitive. Arrange it.
36) P. 14, line 7; react → reacted
37) P. 14, line 8; Concerning “Following optimization, we observed that working at lower concentrations of bifunctional reagents we kinetically obtain the desired ……”, this is an important conclusion description.
First, two “we” are repetitive in a short sentence.
For an example, “Base on the optimization, all the reactions under lower concentrations using the bifunctional precursors (not the reagents!) to successfully produce the desired ……” is an appropriate organic synthesis style.
38) P. 14, line ↑4; (Fig. 5) → (Figure 5) (consistency)
39) P. 15, line 3; Boc chemistry → chemistry using Boc protection protocol
40) P. 15, line 4; the reduction of amide bonds → the reduction of amide groups
41) P. 16, line 1; Two “we” are repetitive in a short sentence. “in which piperazine was incorporated” is better.
42) P. 16, line 2; Two “coupling” are repetitive in a short sentence. The substitution reaction using bromoacetic acid should be provided in the scheme 13 (not Scheme 6).
43) P. 16, line 4; reduction of amide bonds → reduction of amide groups
44) P. 16, line 7; do not → did not
45) P. 16, line ↑4; “Scheme 14” should be moved to the end of the sentence. reduction of amide bonds → reduction of amide groups
46) P. 16, lines ↑2-4; Because two “following” are repetitive, the sentence should be refined.
47) P. 17, the last sentence; Three “use” are repetitive in a short sentence. For example, “were used to” → “were applied to” and “potential use” → “potential development”

The English could be improved to more clearly express the research.
Author Response
Journal: Pharmaceuticals ID: 3286190
Title: " Synthesis of diazacyclic and triazacyclic small molecule libraries using vicinal chiral diamines generated from modified short peptides and their application in drug discovery "
Author(s): Mukund P. Tantak, Ramanjaneyulu Rayala, Prakash Chaudhari, Chhanda C Danta, and Adel Nefzi
Thank you very much for giving us an opportunity to resubmit our manuscript. We really appreciate the kind and extremely useful critiques provided by the reviewer. All points are taken into consideration and have been addressed in the manuscript. The point-by-point responses/changes are described below. We are sure that these changes will make our manuscript significantly better.
The present review manuscript by Nefzi group presents the titled study, that is, synthetic methodology for substituted diaza- and triaza-cyclic compounds. Various these products could be provided by “libraries from libraries” approach mainly as shown in author’s longstanding works (11 original reports). Also, “tea-bag approach” introduced by Dr. Houghten could be applied to the research and development for the project.
Stereochemistry of stereogenic centers is the highlighted topic in synthetic organic chemistry and the chiral discrimination of pharmaceuticals. Accordingly, referring to the original reports, please describe this important subject with the information of the absolute configuration (R/S or D/L amino acids). When the stereochemistry is not strictly confirmed in the equation, asterisks “*” on chiral carbons should be provided in all Schemes and some Figures. Separated equations may provide accessible information to readers.
All libraries were generated from D- and L-amino acids. We only show the stereochemistry for specific individual compounds.
This extensive review manuscript is worthy of note for reasonable members engaged in this area, the reviewer recommends the publication in the journal after major revisions described below.
A number of inappropriate descriptions should be revised along with “comments and suggestions” below.
<General comments and suggestions>
- Title; Two “from” seems to be inappropriate for style of the review. More concise phrase is recommended.
Title modified to: Synthesis of diazacyclic and triazacyclic small molecule libraries using vicinal chiral diamines generated from modified short peptides and their application in drug discovery
- Some “C-”, “N-”, “R”, “S” in the text should be carefully checked and altered to “italic style”
Corrected
- For some long sections, appropriate line breaks will help for smooth reading.
Text is edited to address reviewer comment.
- Titles of Scheme 4 and others; The first words should be capitalized such as synthesis of → Synthesis of (consistency).
Done
- Upper script style is generally used for R1, R2, …, in order to discriminate formulas such as Cn, Nn, etc (lower script). in organic chemistry. But, the manner is not compulsory.
Done for all structures and in the text.
- Heading of Scheme 3 (p. 13) is incorrect and should be changed to Scheme 10. Sequentially, altered the number.
Done. Scheme numbers were corrected.
- In the text, “good yield and purity” and “good yield and purities” are mixed. “good yield with sufficient chemical purity” may be suitable. Arrange this issue.
Done
- All schemes and text sentences; CNBr → BrCN or BrC≡N, COIm2 → Im2CO or Im2C=O, CSIm2 → Im2CS or Im2C=S, (COIm)2 → (ImCO)2 or (ImC=O)2 are appropriate and easy to understand in organic chemistry. But, the manner is not compulsory.
Done. All changes were made in schemes and text.
- In Schemes 8, 9, 11 (not Scheme 4), insertion of “A”, “B”, “C”, … in the arrow equations will support readers’ easier comprehension. In accord with this presentation, it is necessary for the reference in text sentences. But, the manner is not compulsory.
Addressed
<Comments and suggestions>
1) Affiliation of C. C. Danta should be provided.
Done
2) P. 2, lines 5 & 7; “Vicinal amine” and “vicinal diamine” are repetitive. Please check it.
Done
3) P. 4; The abbreviation of “SCLs” (“DCLs” ?) and “KOR” should be explained for readers not specialized in the area.
Done
4) P. 4, line 7; The all D-amino acids tetrapeptide (Difelikefalin) → Difelikefalin, a tetrapeptide with all D-amino acids?
Done
5) P. 7, line 2; Fig 2 → Figure 2 (consistency)
Done
6) P. 7; The abbreviation of “HJs”, “HJ”, “RecG”, and “Mtb” should be explained for readers not specialized in the area.
Done
7) P. 8, line 3; from → form
Corrected
8) P. 8, line ↑3; Fig 3 → Figure 3 (consistency)
Done
9) P. 9, line 9; “Scheme 4” should be to move to line 2 after “was after developed” for readers’ easier comprehension.
Done
10) P. 9, line 6; the vicinal amine → the other vicinal amine
Done
11) P. 9, line ↑3; -3,9- diaza → -3,9-diaza
Done
12) P. 9, line ↑2; dipeptides. As illustrated in Scheme 5, → dipeptides as illustrated in Scheme 5,
Done
13) P. 9, line ↑1; a tertiary nitrogen → a tertiary amino group (consistency) bromoacetyl. → bromoacetyl group.
Done
14) Scheme 5; (COIm)2 → Im2C=O BrCH2COOH → BrCH2COOH
Done
15) Scheme 5; Concerning the formula “N+”, what is the counter anion? Br̶ or Xr̶ ? In order to complete the formula, please provide it.
Done
16) P. 10, line 3; cyclization of an amino and guanidino → cyclization of an amino group with guanidino?
Done
17) P. 10, line 4; guanidine → triazepine?
Done
18) P. 10, line 6; [64-66] should be move to line 2 after “was reported” for readers’ easier comprehension.
Done
19) P. 10, lines 8, 9; Two “from vicinal tetraamines” are repetitive in a short sentence. Arrange it.
Done
20) P. 10, lines 8, 9; Two “Starting from” are repetitive. Arrange it.
Done
21) P. 10, line 11; Concerning “separately treated”, two regioisomers or stereoisomers were produced? If so, please provide the isomer formula in Scheme 7.
Two regioisomers will be formed if there’s no R4 (two primary amines), but it is not the case.
22) P. 10, line 11; were then separately treated in DMF in the presence of ### and DIEA → were then separately treated with ### and DIEA in DMF (as a standard description of synthetic chemistry)
Corrected
23) P. 10, line 12; Concerning “separately stirred in separate vials”, more a smart description should be needed.
Done
24) P. 10, line ↑1; (Fig. 4) → Figure 4 (consistency)
Done
25) P. 11, line 1; Concerning “2065-14”, is this optimized or most active compound? And the correlation of other three compounds should be briefly addressed.
It is our lead compound, and the details are in reference 67. More description was added in the text.
26) P. 11, line 8; “[18]” should be moved after “… compounds” in line 6 for readers’ easier comprehension.
Done
27) P. 11, line ↑7; The N-terminal was N-acylated with → The N-terminal position was acylated to afford the N- precursor with
Done
28) P. 11, line ↑3; thiocarbonylimidazole → ImC=S
Done in the scheme
29) P. 12, line 7; (Scheme 9) should be moved after “was performed” for readers’ easier comprehension.
Done
30) P. 13, line 1; Similarly, and as outlined in Scheme 10, → Similarly, as outlined in Scheme 10,
Done
31) P. 13, line 4; protecting group → group (repetition)
Done
32) Scheme 10 (not 3); What is the condensation reagent in the first equation? POCl3? “Acylation (R2COOH / POCl3)” in all equation is more appropriate in the organic synthesis style.
Corrected and comments are added to the text.
33) P. 14, line 2; tripeptide. → tripeptide [73?].
It is reference 19. Corrected
34) P. 14, line 5; energetically → kinetically
Corrected
35) P. 14, line 6; Two “adjacent” is repetitive. Arrange it.
Done
36) P. 14, line 7; react → reacted
Corrected
37) P. 14, line 8; Concerning “Following optimization, we observed that working at lower concentrations of bifunctional reagents we kinetically obtain the desired ……”, this is an important conclusion description.
Corrected
First, two “we” are repetitive in a short sentence.
Fixed
For an example, “Base on the optimization, all the reactions under lower concentrations using the bifunctional precursors (not the reagents!) to successfully produce the desired ……” is an appropriate organic synthesis style.
Done
38) P. 14, line ↑4; (Fig. 5) → (Figure 5) (consistency)
Done
39) P. 15, line 3; Boc chemistry → chemistry using Boc protection protocol
Corrected
40) P. 15, line 4; the reduction of amide bonds → the reduction of amide groups
Done
41) P. 16, line 1; Two “we” are repetitive in a short sentence. “in which piperazine was incorporated” is better.
Done
42) P. 16, line 2; Two “coupling” are repetitive in a short sentence. The substitution reaction using bromoacetic acid should be provided in the scheme 13 (not Scheme 6).
Corrected
43) P. 16, line 4; reduction of amide bonds → reduction of amide groups
Done
44) P. 16, line 7; do not → did not
Done
45) P. 16, line ↑4; “Scheme 14” should be moved to the end of the sentence. reduction of amide bonds → reduction of amide groups
Done
46) P. 16, lines ↑2-4; Because two “following” are repetitive, the sentence should be refined.
Corrected
47) P. 17, the last sentence; Three “use” are repetitive in a short sentence. For example, “were used to” → “were applied to” and “potential use” → “potential development”
Corrected

Reviewer 2 Report
Comments and Suggestions for Authors
It is not clear from the article at first whether this is a review or a research article. Only at the end of the article do you read that all the authors agree with this perspective. This review has a high percentage of overlap with other sources (63%).
This review describes the use of vicinal diamines obtained from modified short peptides to access substituted diaza- and triazacyclic compounds.
Nitrogen heterocycles are of great importance in the pharmaceutical field, since these compounds are found in numerous bioactive structures.
Their synthesis often requires several steps or the use of functionalized starting materials.
The review describes examples of the synthesis of diazacyclic and triazacyclic small molecule libraries from vicinal chiral polyamines obtained from modified short peptides and their application for the identification of highly active compounds.
Small molecule diaza- and triazacyclic compounds with various substitution patterns and embedded in various molecular frameworks are important structural classes in the search for bioactivity.
The review is very difficult to read, as there is no division into subtopics, not even an introduction, let alone other subheadings. The review needs to be rewritten and structured according to the stated directions: screening of diazacyclic and triazacyclic libraries has shown the usefulness of these classes of compounds for de novo identification of highly active compounds, including antimalarial, antimicrobial, antifibrotic, potent analgesics and antitumor agents.
Author Response
Journal: Pharmaceuticals ID: 3286190
Title: " Synthesis of diazacyclic and triazacyclic small molecule libraries using vicinal chiral diamines generated from modified short peptides and their application in drug discovery "
Author(s): Mukund P. Tantak, Ramanjaneyulu Rayala, Prakash Chaudhari, Chhanda C Danta, and Adel Nefzi
Thank you very much for giving us an opportunity to resubmit our manuscript. We really appreciate the kind and extremely useful critiques provided by the reviewer. All points are taken into consideration and have been addressed in the manuscript. The point-by-point responses/changes are described below. We are sure that these changes will make our manuscript significantly better.
It is not clear from the article at first whether this is a review or a research article. Only at the end of the article do you read that all the authors agree with this perspective. This review has a high percentage of overlap with other sources (63%).
It is a review. We are the only laboratory working on the generation of heterocyclic peptidomimetics derived from the exhaustive reduction of polyamides (chiral polyvicinal maines).
This review describes the use of vicinal diamines obtained from modified short peptides to access substituted diaza- and triazacyclic compounds.
Nitrogen heterocycles are of great importance in the pharmaceutical field, since these compounds are found in numerous bioactive structures.
Their synthesis often requires several steps or the use of functionalized starting materials.
The review describes examples of the synthesis of diazacyclic and triazacyclic small molecule libraries from vicinal chiral polyamines obtained from modified short peptides and their application for the identification of highly active compounds.
Small molecule diaza- and triazacyclic compounds with various substitution patterns and embedded in various molecular frameworks are important structural classes in the search for bioactivity.
The review is very difficult to read, as there is no division into subtopics, not even an introduction, let alone other subheadings. The review needs to be rewritten and structured according to the stated directions: screening of diazacyclic and triazacyclic libraries has shown the usefulness of these classes of compounds for de novo identification of highly active compounds, including antimalarial, antimicrobial, antifibrotic, potent analgesics and antitumor agents.
The review was edited and rewritten. Per reviewer suggestion, the paper was divided into subtopics, and subheadings were added.
More data about the identification of active compounds were added.

Reviewer 3 Report
Comments and Suggestions for Authors
The work titled "Synthesis of diazacyclic and triazacyclic small molecule libraries from vicinal chiral diamines generated from modified short peptides and their application in drug discovery” presents valuable insights into synthesis of small molecules base on diazacyclic and triazacyclic and their application in drug discovery. The review is interesting and could offers significant contributions to the field of developing new therapeutics. The work presents very interesting review of synthesis of the title compound. The paper is well written. My opinion is that the topic is of interest to the readership of Pharmaceuticals so the manuscript should be accepted, but after minor revision. Below are some comments and suggestions for improving the manuscript:
1. The article contains primarily an overview of synthesis methods of diazacyclic and triazacyclic small molecules. Less focus is given to the application side of products. In my opinion, this topic should be described in more detail. Otherwise, the title of the article should be changed as it does not reflect the content.
2. The authors should also work on editing the article. There is no summary. Schemes 3, 4, 5, 6 and 7 appear twice in the article, even though they are different schemes.
Author Response
Journal: Pharmaceuticals ID: 3286190
Title: " Synthesis of diazacyclic and triazacyclic small molecule libraries using vicinal chiral diamines generated from modified short peptides and their application in drug discovery "
Author(s): Mukund P. Tantak, Ramanjaneyulu Rayala, Prakash Chaudhari, Chhanda C Danta, and Adel Nefzi
Thank you very much for giving us an opportunity to resubmit our manuscript. We really appreciate the kind and extremely useful critiques provided by the reviewer. All points are taken into consideration and have been addressed in the manuscript. The point-by-point responses/changes are described below. We are sure that these changes will make our manuscript significantly better.
The work titled "Synthesis of diazacyclic and triazacyclic small molecule libraries from vicinal chiral diamines generated from modified short peptides and their application in drug discovery” presents valuable insights into synthesis of small molecules base on diazacyclic and triazacyclic and their application in drug discovery. The review is interesting and could offers significant contributions to the field of developing new therapeutics. The work presents very interesting review of synthesis of the title compound. The paper is well written. My opinion is that the topic is of interest to the readership of Pharmaceuticals so the manuscript should be accepted, but after minor revision. Below are some comments and suggestions for improving the manuscript:
- The article contains primarily an overview of synthesis methods of diazacyclic and triazacyclic small molecules. Less focus is given to the application side of products. In my opinion, this topic should be described in more detail. Otherwise, the title of the article should be changed as it does not reflect the content.
More biological data were added to the text.
- The authors should also work on editing the article. There is no summary. Schemes 3, 4, 5, 6 and 7 appear twice in the article, even though they are different schemes.
The paper is edited and the numbers for the schemes were corrected.

Round 2
Reviewer 1 Report
Comments and Suggestions for Authors
The reviewer has checked this revised review manuscript by Nefzi group. The only one request is that, as the first review pointed out, the following recommendation is re-considered.
<Stereochemistry of stereogenic centers is the highlighted topic in synthetic organic chemistry and the chiral discrimination of pharmaceuticals. Accordingly, referring to the original reports, please describe this important subject with the information of the absolute configuration (R/S or D/L amino acids). When the stereochemistry is not strictly confirmed in the equation, asterisks “*” on chiral carbons should be provided in all Schemes and some Figures. Separated equations may provide the accessible information to readers.>
Please reconsider this comment/suggestion.
Comments on the Quality of English Language
The English could be improved to more clearly express the research.
Author Response
Thank you very much for giving us an opportunity to resubmit our manuscript. We really appreciate the kind and extremely useful critiques provided by the reviewer. All points are taken into consideration and have been addressed in the manuscript. The point-by-point responses/changes are described below. We are sure that these changes will make our manuscript significantly better.
Most of the chemistry presented in the paper applies to L- and D-amino acids.
We assigned the stereochemistry in all schemes and Figures in which the building blocks are defined such as in Figures 2-5, and the use of L-lysine or L-proline as spacers in Schemes 9, 10, 12 and 14.
Thank you
Reviewer 2 Report
Comments and Suggestions for Authors
I have no comments.
Author Response
Thank you very much for giving us an opportunity to resubmit our manuscript. We really appreciate the kind and extremely useful critiques provided by the reviewer. All points are taken into consideration and have been addressed in the manuscript. The point-by-point responses/changes are described below. We are sure that these changes will make our manuscript significantly better.
The reviewer has no comments.
Thank you